# Extrahepatic Recurrence After Surgical Resection of Hepatocellular Carcinoma Without Intrahepatic Recurrence: A Multi-Institutional Observational Study

**DOI:** 10.3390/cancers17091417

**Published:** 2025-04-23

**Authors:** Ga Ram You, Shin Young Park, Su Hyeon Cho, Sung Bum Cho, Yang Seok Koh, Chang Hun Lee, Hoon Gil Jo, Sung Kyu Choi, Jae Hyun Yoon

**Affiliations:** 1Department of Gastroenterology and Hepatology, Chonnam National University Hwasun Hospital and Medical School, Hwasun 58128, Republic of Korea; rugaram27@hanmail.net (G.R.Y.); portalvein@naver.com (S.B.C.); 2Department of Gastroenterology and Hepatology, Chonnam National University Hospital and Medical School, Gwangju 61469, Republic of Korea; breezy87@naver.com (S.Y.P.); hyekyo0122@naver.com (S.H.C.); choisk@jnu.ac.kr (S.K.C.); 3Department of Surgery, Chonnam National University Hwasun Hospital and Medical School, Hwasun 58128, Republic of Korea; yskoh@chonnam.ac.kr; 4Department of Gastroenterology and Hepatology, Jeonbuk National University Hospital and Medical School, Jeonju 54907, Republic of Korea; chleemd@jbnu.ac.kr; 5Department of Gastroenterology and Hepatology, Wonkwang University Hospital and Medical School, Iksan 54538, Republic of Korea; jojo420600@gmail.com

**Keywords:** extrahepatic recurrence, HCC, prognosis, surgery, survival

## Abstract

Hepatocellular carcinoma (HCC) is the most common type of liver cancer, and its recurrence outside the liver, known as extrahepatic recurrence (EHR), is associated with poor prognosis. While EHR typically occurs in patients with high-risk factors, its development without detectable HCC remaining in the liver is not well understood. This study aims to identify the clinical characteristics and risk factors associated with EHR in patients who have undergone curative liver surgery for HCC. By analyzing data from 569 patients, we found that microvascular invasion, tumor necrosis, and advanced tumor stage significantly increased the likelihood of EHR. Furthermore, EHR was linked to shorter survival after surgery. These findings highlight the importance of close monitoring in high-risk patients, as detecting and managing EHR early may improve outcomes.

## 1. Introduction

Hepatocellular carcinoma (HCC) is the third leading cause of cancer-related mortality and the sixth most prevalent cancer worldwide, with its incidence rapidly increasing [1]. Despite therapeutic advancements, managing HCC remains challenging due to its aggressive nature and high recurrence rates [2,3]. Surgical resection, liver transplantation, and radiofrequency ablation (RFA) are key treatments for early-stage HCC, with surgical resection offering a potentially curative option for eligible patients. However, despite advancements in medical devices and surgical techniques, postoperative recurrence remains a significant challenge in HCC surgery. Studies report recurrence rates of up to 43.0% within 2 years post-surgery [4,5], while our previous research shows 3-year and 5-year recurrence rates of 50.5% and 63.0%, respectively [6].

Previous research reports that 38.3% of patients with extrahepatic recurrence (EHR) have no intrahepatic recurrence (IHR) at EHR diagnosis; however, comprehensive data on these patients remains limited [7]. Since IHR is the most common recurrence form in HCC [8], treatment predominantly relies on locoregional therapies such as transarterial chemotherapy, selective internal radiation therapy, and thermal ablative therapy, but EHR development is linked to poorer survival outcomes [9,10]. Although various locoregional and systemic therapies are available for managing IHR, treatment options for addressing EHR remain limited. Predicting EHR without IHR following surgery is particularly challenging, highlighting the need for predictive models and risk assessment tools [11,12].

Therefore, this study aims to comprehensively investigate the clinical characteristics and risk factors associated with EHR without IHR following surgical resection and their influence on survival outcomes. To address this, we developed a parametric model to predict EHR occurrence without IHR post-surgery. Through a detailed review and analysis, this study could clarify the current understanding of EHR without IHR following curative HCC resection, identify predictive factors, and propose strategies for enhanced surveillance and management.

## 2. Materials and Methods

### 2.1. Patients

All patients who underwent surgical resection as first-line therapy for HCC between January 2004 and December 2019 at four tertiary hospitals were included in this study. As the rupture of HCC itself may represent a significant risk factor for the development of EHR, patients whose initial diagnosis of HCC was associated with hepatic rupture were excluded. Of the 1105 patients, those with non-HCC tumors, prior HCC treatment, or incomplete data were excluded (Figure 1), leaving 1066 patients for analysis. Baseline characteristics at the time of HCC diagnosis was examined by grouping patients based on their IHR and/or EHR status (Appendix A). Surgical findings for each group, as well as factors associated with EHR in the entire cohort, were also assessed (Appendix A). To analyze factors associated with EHR without prior IHR, we excluded patients who developed IHR before EHR, resulting in the inclusion of 569 patients for subsequent analysis. Surgical resection was performed by treating physicians based on tumor stage, liver function, and the physical condition of the patients. To minimize surgical technique-related bias and undetected extrahepatic metastases before resection, patients who developed EHR within 60 days post-surgery were excluded. While a longer exclusion period (e.g., 6–12 months) may further reduce the possibility of including pre-existing, undetected metastases, we adopted a 60-day threshold in consideration of the limited number of eligible patients with EHR without IHR. This time frame was chosen to strike a balance between specificity and statistical power.

### 2.2. Baseline HCC Staging, Surgical Resection, and Follow-Up

HCC diagnosis followed the guidelines of the Korean Liver Cancer Study Group and the National Cancer Center [13]. Staging at diagnosis utilized the modified Union for International Cancer Control (mUICC) [14] and Barcelona Clinic Liver Cancer (BCLC) classification systems [15]. The Milan criteria, defined by radiological findings at initial diagnosis, were used to guide therapeutic decisions [16]. Routine evaluations, including abdominal computed tomography (CT), magnetic resonance imaging (MRI), and serological tumor marker assessments, were conducted 1 month post-resection and every 3–6 months thereafter.

Tumor size was assessed using CT and MRI. HCC histological grading followed the Edmondson and Steiner criteria [17]. Macrovascular invasion was identified through vascular encroachment visible on CT or MRI, while microvascular invasion was determined by examining resected tumor vasculature microscopically. Pathological analysis of the specimens confirmed lymph node metastasis, serosal and bile duct invasion, capsule formation, multicentricity, satellite nodules, intrahepatic metastasis, tumor necrosis, hemorrhage, and fatty transformation. Overall survival was defined as the time from HCC surgical resection to death or last follow-up.

### 2.3. Diagnosis of Recurrence and EHR

EHR was confirmed through contrast-enhanced CT or MRI. Pathological examinations were conducted at the discretion of the consulting physician. Additional diagnostics, including chest radiography, positron emission tomography–computed tomography (PET–CT), brain MRI or CT, and bone scintigraphy, were conducted when symptoms suggested tumor metastasis. While most EHR cases were identified during routine follow-up, a few were detected through the assessment of new symptoms or significant increases in alpha-fetoprotein (AFP) levels or serial AFP elevation without clear intrahepatic lesions.

### 2.4. Ethics Statement

This study was approved by the institutional review boards of the participating institutions (Institutional Review Board of Chonnam National University Hospital, No. CNUH-2019-203). Owing to the retrospective design and the use of de-identified data, the requirement for informed consent was waived. The study was conducted in accordance with the principles of the Declaration of Helsinki (1975).

### 2.5. Statistical Analysis

Data are presented as means ± standard deviations or medians with ranges, depending on the data distribution. Analysis of variance was employed to identify statistically significant differences among the means of three or more groups. Univariate analyses were conducted with the chi-squared test or Student’s *t*-test, as applicable. Variables with *p*-values < 0.05 in univariate analyses were further analyzed using multivariate logistic regression to identify predictors of EHR. A multivariable Cox regression model was developed using a stepwise backward selection method, retaining variables with *p*-values < 0.05 as predictors. EHR risk levels were classified based on the presence of risk factors, and Kaplan–Meier survival curves were generated for each risk category. All statistical analyses were conducted using SPSS software version 27.0 (IBM Corp., Armonk, NY, USA).

## 3. Results

### 3.1. Baseline Characteristics of the Enrolled Patients

The baseline characteristics of the enrolled patients were analyzed (Appendix A). Of the 569 patients, 373 had IHR without EHR, while 124 had IHR and EHR, either simultaneously or subsequently. Baseline characteristics were reassessed after excluding patients with IHR (Table 1). Among the 569 patients, 38 (6.7%) developed EHR without IHR, while 531 (93.3%) had no HCC recurrence following surgical resection. The median follow-up for patients with EHR without IHR was 3.03 years; during this period, 26 (68.54%) of the 38 patients died. Conversely, the median follow-up for the 531 patients without recurrence was 4.08 years, with 32 (6.0%) deaths. Comparative analysis indicated that patients with EHR without IHR exhibited higher rates of hepatitis C virus infection, elevated prothrombin induced by vitamin K absence (PIVKA)-II levels, larger tumors, and more tumors. They also had a higher prevalence of advanced tumor stages, including pathological mUICC stage (≥III), BCLC stage (≥C), cases exceeding the Milan criteria, and a higher incidence of macrovascular invasion.

### 3.2. Comparison of Clinical and Pathological Findings

Clinical and pathological findings were compared between patients with EHR without IHR and those without recurrence (Table 2). Patients with EHR exhibited significantly higher rates of microvascular invasion (31.6% vs. 7.8%, *p* < 0.001), satellite nodules (23.7% vs. 8.0%, *p* = 0.001), tumor necrosis (65.8% vs. 29.3%, *p* < 0.001), hemorrhage (57.9% vs. 32.4%, *p* = 0.001), and fatty change (33.8% vs. 16.2%, *p* = 0.027). Additionally, the pseudoglandular histological type was more common in patients with EHR (52.6% vs. 32.7%, *p* = 0.012).

### 3.3. Clinical Findings at the First Recurrence of Patients with EHR Without IHR

Clinical findings at first recurrence in patients with EHR and no intrahepatic HCC after surgical resection were analyzed (Appendix A). The median recurrence-free survival (RFS) was 1.07 years. The lungs were the most frequent metastatic site (n = 15, 36.59%), with multiple nodules in 26.83% of cases. Other common metastatic sites included the lymph nodes (n = 7, 17.07%), bone (n = 7, 17.07%), peritoneum (n = 5, 12.20%), adrenal glands (n = 5, 12.20%), and brain (n = 2, 4.88%). Abdominal contrast-enhanced CT identified the most metastases (56.41%), followed by chest-enhanced CT (17.95%) and PET–CT (10.26%). Four patients experienced multiple EHR, with the following combinations: lymph nodes and peritoneum, lung nodules and brain, lung nodules and lymph nodes, and bone and adrenal glands.

### 3.4. Analysis of Factors Associated with EHR

Cox regression analysis was conducted to identify factors associated with the development of EHR without IHR (Table 3). The multivariate analysis revealed the following factors as significantly associated with EHR: microvascular invasion (hazard ratio [HR] = 2.418; 95% confidence interval [CI], 1.146–5.099, *p* = 0.020), tumor necrosis (HR = 2.592; 95% CI, 1.264–5.316, *p* = 0.009), and an initial tumor stage exceeding the Milan criteria (HR = 3.008; 95% CI, 1.548–5.843, *p* = 0.001).

### 3.5. Cumulative Rates of EHR Categorized Based on the Number of Risk Factors

Among the 569 enrolled patients, 38 (6.7%) developed EHR over a median follow-up of 3.03 years (Figure 1). The median time to EHR was 1.04 years. The cumulative incidences of EHR without IHR at 1, 3, 5, and 10 years were 3.3%, 6.3%, 7.7%, and 9.8%, respectively (Figure 2). Patients were stratified based on the three risk factors identified in the multivariate analysis. The EHR rate increased with the number of risk factors (Figure 3).

### 3.6. Analysis of Factors Associated with Overall Survival After Surgical Resection

A Cox regression analysis was conducted to identify factors associated with overall survival following HCC surgical resection (Table 4). Of the 13 risk factors identified in the univariate analysis, the multivariate analysis revealed two statistically significant factors: multiple tumors (HR = 3.873; 95% CI, 1.988–7.546; *p* < 0.001) and EHR development (HR = 13.814; 95% CI, 7.971–23.938; *p* < 0.001).

Overall survival rates were compared based on EHR development, showing significant differences between the two groups (Figure 4a). For the non-EHR group, the cumulative overall survival rates at 1, 3, 5, and 10 years were 99.2%, 96.6%, 94.4%, and 82.0%, respectively. Conversely, the survival rates for the EHR group at the same time points were 94.6%, 62.5%, 36.9%, and 16.8% (*p* < 0.001). Furthermore, overall survival was analyzed according to the number of risk factors. A decline in overall survival was observed as the number of risk factors increased. Patients were categorized into two groups based on their number of risk factors: low-risk (0–1 risk factor) and high-risk (2–3 risk factors). Significant differences were observed in cumulative EHR and overall survival rates (*p* < 0.001, Appendix A). The cumulative EHR rates at 1, 3, 5, and 10 years were 1.7%, 3.1%, 3.9%, and 5.1%, respectively, for the low-risk group, and 12.1%, 23.9%, 29.1%, and 36.5%, respectively, for the high-risk group. The overall survival rates for the low-risk group at 1, 3, 5, and 10 years were 99.3%, 95.5%, 93.2%, and 77.2%, respectively. For the high-risk group, the survival rates were 97.6%, 85.4%, 69.1%, and 62.6%, respectively. The overall survival rates, categorized by IHR and/or EHR development, showed that the group with no recurrence exhibited the highest survival rates, significantly surpassing all other groups. Patients with IHR without EHR had better survival rates than those with EHR (*p* < 0.001). No significant difference in survival rates was observed between the EHR-only group and the EHR with IHR group (Appendix A).

## 4. Discussion

In this study, we examined EHR without IHR following curative surgical resection for HCC, using data from multiple high-volume medical centers. The goal is to identify predictive factors and develop improved strategies for surveillance and management. EHR without IHR is a sudden and challenging condition for clinicians and patients managing HCC; however, it remains understudied. Our findings revealed that 6.7% of patients (38 patients) developed EHR without IHR, emphasizing the importance of considering EHR even in the absence of IHR.

EHR development in patients with HCC correlates with high tumor burden and an aggressive tumor phenotype, resulting in poor clinical outcomes [18]. Yoon et al. report poor survival outcomes in patients with EHR after RFA or surgical resection for HCC [6,19]. An observational study of 779 patients reveals that early EHR, occurring within 2 years of surgical resection, is linked to significantly worse survival outcomes (HR 6.77) over 15 years [10]. Although EHR generally has a poor prognosis, favorable outcomes have been observed in selected patients with limited EHR who undergo treatments such as metastasectomy [20,21]. Therefore, accurately predicting the risk of EHR is crucial for improving patient outcomes. Identifying individuals at high risk and implementing close follow-up surveillance enables the early detection of EHR and timely postoperative adjuvant therapy.

In this study, 6.7% of patients who underwent surgical resection developed EHR without recurrence of HCC. This subgroup may represent those who experience sudden EHR of HCC during postoperative follow-up, excluding those with prior IHR or EHR following IHR. The cumulative EHR rates at 1, 5, and 10 years were 5.2%, 7.7%, and 9.8%, respectively. This significant increase underscores the need for clinicians to focus on monitoring and managing EHR in patients. Cox regression analyses (univariate and multivariate) were conducted to assess the risk factors for EHR (Table 3). In the multivariate model, microvascular invasion, tumor necrosis, and tumor stage beyond the Milan criteria emerged as statistically significant factors associated with EHR occurrence. Stratifying patients by the number of these risk factors revealed a stepwise increase in cumulative EHR rates with an increasing number of risk factors (Figure 3). Microvascular invasion in HCC is a well-established prognostic factor after HCC treatments, including surgical resection [22,23], and is linked to EHR development in other studies. Wei T et al. analyzed 635 patients who underwent HCC surgical resection and developed a nomogram for EHR prediction, identifying microvascular invasion as a significant risk factor [24]. Similarly, Wei HW et al. evaluated 227 patients to create a nomogram model for EHR prediction, with multivariate analysis also identifying microvascular invasion as a significant factor [25]. Tumor necrosis was another significant factor associated with EHR in this study. Consistent with these findings, Wei T et al. report that tumor necrosis correlates with overall survival, RFS, and advanced tumor features in patients undergoing HCC surgery [26]. Ling et al. also reveal tumor necrosis as a potential marker linked to HCC aggressiveness [27]. The Milan criteria, originally proposed to identify liver transplantation (LT) candidates with a high RFS rate [16], serve as a surrogate marker for recurrence in other treatment modalities, including surgical resection and transarterial chemoembolization [28]. Andreou et al. examined predictive factors for EHR post-LT, revealing that tumors exceeding the Milan criteria significantly correlate with EHR in multivariate analysis.

In this context, the potential role of pre-transplant biopsy in evaluating tumor biology, such as DNA ploidy or DNA index, has also been highlighted as a way to improve risk stratification in LT candidates. Favorable tumor biology, as indicated by these molecular features, may help predict a lower risk of extrahepatic recurrence and refine selection criteria for transplantation [11,29].

Consistent with these findings, our study also reveals a strong association between EHR and tumor stage beyond the Milan criteria [30]. Patients were reclassified based on the presence of three risk factors, with cumulative EHR rates increasing progressively as the number of risk factors increased (Figure 3).

Furthermore, the effect of EHR on patient survival following surgical resection is significant. When patients were categorized by the presence or absence of EHR, overall survival rates differed significantly (Figure 4a), with a 5-year survival rate of 94.4% for those without EHR compared to 36.9% for those with EHR. In the multivariate analysis of risk factors for poor overall survival, EHR was the most significant factor (HR = 12.838, *p* < 0.001), followed by the presence of multiple tumors (HR = 3.720, *p* < 0.001) and MELD ≥ 10 (HR = 3.315, *p* = 0.005) (Table 4). Multiple tumors, a recognized prognostic factor for HCC, are included in most tumor staging systems [14,31] and are also associated with EHR [24].

When applying the identified risk factors for EHR to overall survival rates, survival declined exponentially as the number of risk factors increased (Figure 4b).

To apply these risk factors in clinical practice, patients were categorized into low-risk and high-risk groups based on their number of risk factors (0–1: low risk, 2–3: high risk). A significant difference in cumulative EHR and overall survival rates was observed between the two groups (Appendix A, *p* < 0.001). Certain locations showed higher frequencies of EHR development, though these locations varied (Appendix A). Approximately 41.1% of patients were diagnosed with EHR using imaging modalities other than abdominal CT or MRI, with a median time to EHR occurrence of 1.07 years. Although tumor markers (serum AFP or PIVKA-II levels) were elevated in many patients, no significant correlation was found in multivariate analysis. Among the tumor markers evaluated, various AFP thresholds were considered, and a cut-off value of 1000 ng/mL was selected based on its strongest association with EHR in our dataset (AUC 0.591). This threshold is also supported by previous studies reporting poor prognosis and an increased risk of metastasis in patients with markedly elevated AFP levels.

PIVKA-II (DCP), although a well-known marker of HCC, was excluded from the multivariate analysis due to a substantial proportion of missing data, which limited its utility as a reliable variable in our statistical model and is acknowledged as a limitation of this study. Integrating the novel risk factors identified in this study could help clinicians assess individual EHR risk and enhance early detection.

While patients with EHR after treatment of HCC generally have limited therapeutic options beyond systemic therapy, potentially curative interventions such as metastasectomy or radiotherapy may be considered in select instances, particularly when EHR presents as a solitary lesion or involves a limited number of sites (oligometastasis), and have been linked to improved overall survival [32,33,34,35]. However, in the majority of EHR cases, multiple metastases are more common than oligometastatic presentations, which contributes to less favorable clinical outcomes. In this context, we aimed to identify potential risk factors associated with EHR in the absence of IHR, to facilitate early detection and timely intervention. Unfortunately, there remains no standardized imaging modality or surveillance interval specifically tailored for the detection of EHR following the surgical resection of HCC. Nonetheless, in our cohort, 56.41% of EHR cases were identified using contrast-enhanced abdominal CT and 17.95% via chest CT.

Furthermore, immunosuppression and patient-related factors such as age, tobacco and alcohol use, and the underlying etiology of liver disease have also been associated with an increased risk of de novo extrahepatic malignancies, particularly in the post-transplant population. These factors may compromise immune surveillance, facilitate oncogenic viral reactivation, and contribute to a proinflammatory microenvironment, thereby increasing malignancy risk and leading to poorer outcomes. As highlighted by Choudhary et al., identifying and managing these risk factors through tailored surveillance strategies can enhance early detection and potentially improve long-term prognosis in this high-risk cohort [36].

This study has some limitations. First, its retrospective design depends on patient medical records, which may introduce bias. Additionally, the lack of a standardized treatment protocol or resection criteria, along with inconsistent post-treatment surveillance schedules, represents a significant limitation. Surveillance modalities were determined at the discretion of the treating physician, contributing to variability. To address these limitations, we enrolled a large cohort from four high-volume centers and excluded individuals with prior IHR or concomitant EHR with IHR, reducing potential bias from differing treatment modalities following HCC recurrence. Furthermore, EHR diagnosis primarily relies on medical imaging, which may miss occult primary malignancies outside the liver. In cases with an unclear tumor origin, pathological confirmation was conducted at the discretion of the treating physician. Also, microvascular invasion and tumor necrosis can only be confirmed through pathological examination following surgical resection. However, given that surgical resection remains one of the most definitive curative treatments for HCC, particularly in its early stages, and that no established neoadjuvant treatment exists for patients at high risk of post-surgical recurrence, identifying post-surgical pathological features associated with the risk of EHR may provide meaningful insights for clinical practice.

The decision to use a 60-day threshold for excluding early recurrence was intentionally chosen to enhance specificity; nevertheless, we acknowledge that this cut-off remains somewhat arbitrary. Future large-scale studies focusing on patients with no recurrence for a prolonged period (e.g., one year or longer) after surgery would help refine the distinction between pre-existing and true postoperative recurrence.

## 5. Conclusions

In conclusion, this study underscores the importance of monitoring and managing elevated EHR markers in patients undergoing surgical resection for HCC, even in the absence of IHR. A risk stratification approach based on these factors facilitates a personalized strategy for early detection and intervention, potentially improving overall survival rates.

## Figures and Tables

**Figure 1 cancers-17-01417-f001:**
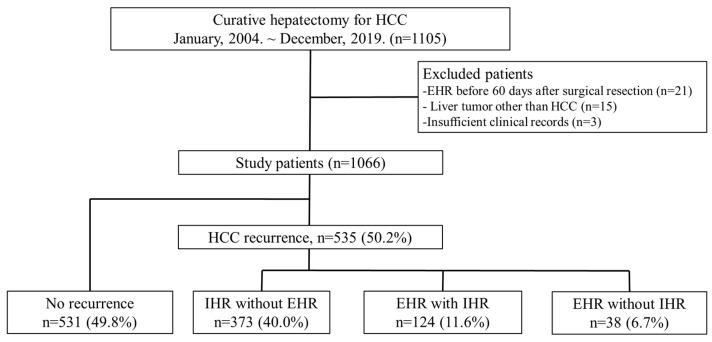
Flowchart depicting patient enrollment in this study. Abbreviations: IHR, intrahepatic recurrence; EHR, extrahepatic recurrence; HCC, hepatocellular carcinoma.

**Figure 2 cancers-17-01417-f002:**
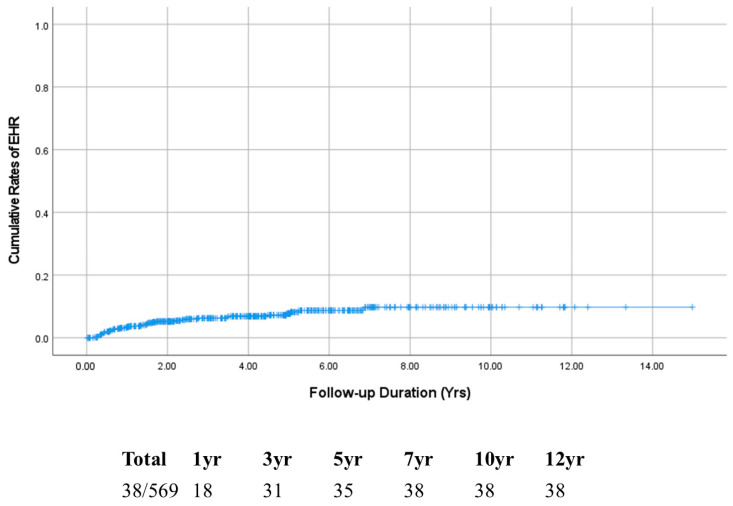
Cumulative EHR rates after surgical resection of HCC. Abbreviations: EHR, extrahepatic recurrence; HCC, hepatocellular carcinoma.

**Figure 3 cancers-17-01417-f003:**
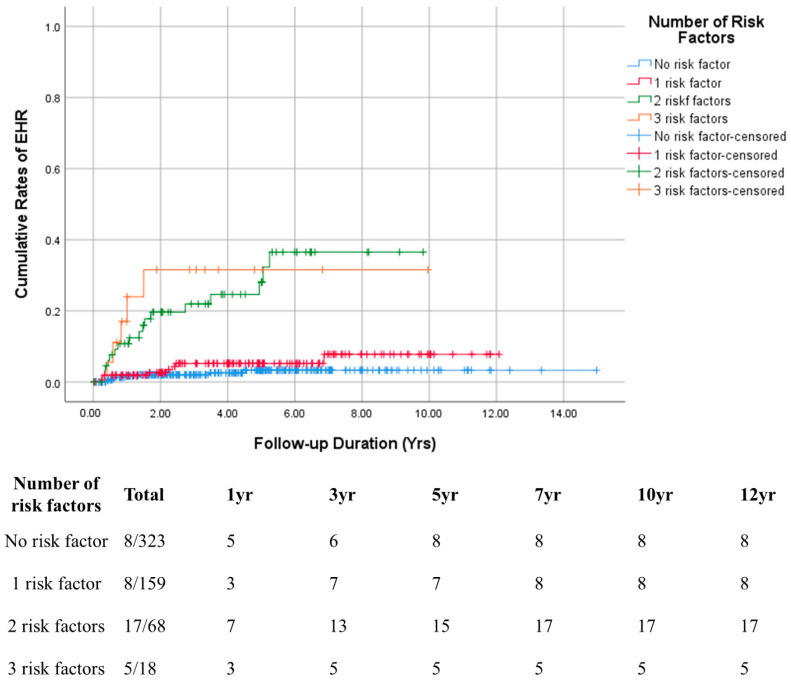
Cumulative EHR rates categorized based on the number of risk factors. Abbreviations: EHR, extrahepatic recurrence.

**Figure 4 cancers-17-01417-f004:**
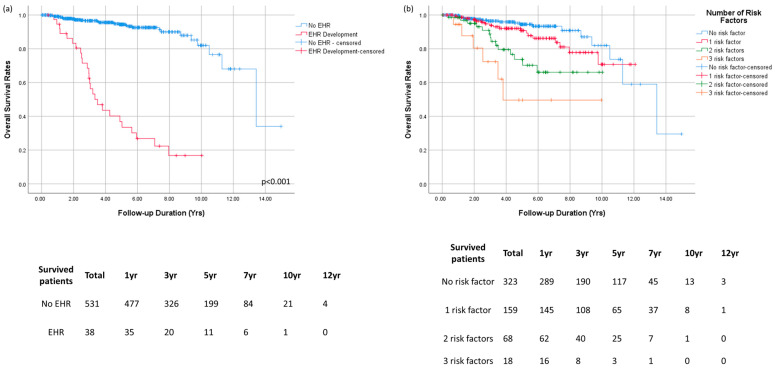
(**a**) Overall survival rates between patient groups with and without EHR; (**b**) Patient overall survival rates after surgical resection of HCC, categorized by the number of risk factors. Abbreviations: EHR, extrahepatic recurrence; HCC, hepatocellular carcinoma.

**Table 1 cancers-17-01417-t001:** Baseline characteristics of patients (n = 569) at initial HCC diagnosis.

Characteristic	Patients Without Recurrence (n = 531)	Patients Without Intrahepatic ^1^ HCC at EHR (n = 38)	*p*-Value
Age (years)	59.87 ± 10.22	58.46 ± 9.72	0.412
Male (n, %)	437 (82.3)	34 (89.5)	0.258
BMI (kg/m^2^)	24.28 ± 3.16	23.65 ± 2.65	0.228
Etiology of liver cirrhosis, n (%)			
HBV	321 (74.7)	26 (76.5)	0.814
HCV	25 (5.8)	5 (14.7)	0.042
Alcoholism	85 (19.8)	3 (8.8)	0.117
Child–Pugh Score, n (%) A/B	504 (99.0)/5 (1.0)	37 (97.4)/1 (2.6)	0.346
ALBI grade ≥ 2, n (%)	56 (10.6)	7 (19.4)	0.103
ALP (U/L)	89.66 ± 47.04	91.84 ± 41.90	0.781
Albumin (mg/dL)	4.37 ± 0.41	4.27 ± 0.41	0.149
Cr (mg/dL)	0.93 ± 0.44	0.97 ± 0.18	0.567
Serum AFP (IU/mL)	911.93 ± 7599.14	2273.49 ± 8293.22	0.296
PIVKA-II (mAU/mL)	713.57 ± 3170.28	5165.72 ± 12,041.60	<0.001
AST (IU/mL)	36.68 ± 26.63	40.82 ± 26.89	0.356
ALT (IU/mL)	34.23 ± 29.42	39.97 ± 33.03	0.249
Sum of tumor size (cm)	3.78 ± 2.27	6.10 ± 3.84	<0.001
Tumor numbers	1.08 ± 0.36	1.29 ± 0.84	0.003
Pathological mUICC stage (≥III), n (%)	61 (11.5)	13 (34.2)	<0.001
BCLC stage (≥C), n (%)	16 (3.0)	5 (13.2)	0.001
Beyond the Milan criteria, n (%)	98 (18.5)	20 (52.6)	<0.001
Macrovascular invasion, n (%)	35 (6.6)	5 (13.2)	0.130

Values are presented as mean ± SD. ^1^ Abbreviations: SD, standard deviation; HCC, hepatocellular carcinoma; EHR, extrahepatic recurrence; BMI, body mass index; HBV, hepatitis B virus; HCV, hepatitis C virus; ALBI, albumin–bilirubin; ALP, alkaline phosphatase; Cr, creatinine; AFP, alpha-fetoprotein; PIVKA-II, prothrombin induced by vitamin K absence-II; AST, aspartate transaminase; ALT, alanine transaminase; mUICC, modified Union for International Cancer Control; BCLC, Barcelona Clinic Liver Cancer.

**Table 2 cancers-17-01417-t002:** Surgical findings of patients at initial HCC diagnosis (n = 569).

Characteristic	Patients Without Recurrence (n = 531)	Patients Without Intrahepatic ^1^ HCC at EHR (n = 38)	*p*-Value
Margin involvement, n (%)	14 (5.0)	1 (3.8)	0.797
Microvascular invasion, n (%)	41 (7.8)	12 (31.6)	<0.001
Serosal invasion, n (%)	10 (1.9)	2(5.3)	0.171
Bile duct invasion, n (%)	5 (1.0)	0 (0.0)	0.544
Capsule formation, n (%)	350 (67.3)	26 (68.4)	0.875
Multicentricity, n (%)	25 (4.8)	3 (7.9)	0.393
Satellite nodule, n (%)	42 (8.0)	9 (23.7)	0.001
Necrosis, n (%)	153 (29.3)	25 (65.8)	<0.001
Hemorrhage, n (%)	169 (32.4)	22 (57.9)	0.001
Fatty change, n (%)	176 (33.8)	6 (16.2)	0.027
Cell type, n (%)			
Clear type	84 (16.2)	6(15.8)	0.945
Hepatic type	488 (94.2)	36 (94.7)	0.893
Classic type	326 (62.9)	28 (73.7)	0.184
Major Edmondson–Steiner grade ≥ 3, n (%)	186 (35.2)	20 (52.6)	0.031
Worst Edmondson–Steiner grade ≥ 3, n (%)	356 (67.3)	34 (89.5)	0.004
Histologic type, n (%)			
Trabecular type	474 (91.7)	36 (94.7)	0.506
Pseudoglandular type	169 (32.7)	20 (52.6)	0.012
Acinar type	10 (1.9)	0 (0.0)	0.387
Compact type	51 (9.9)	4 (10.5)	0.895
Solid type	34 (6.6)	3 (7.9)	0.753

^1^ Abbreviations: HCC, hepatocellular carcinoma; EHR, extrahepatic recurrence.

**Table 3 cancers-17-01417-t003:** Factors associated with the absence of intrahepatic HCC at EHR diagnosis.

Factors	Univariate Analysis		Multivariate Analysis	
	^1^ HR (95% CI)	*p*-Value	HR (95% CI)	*p*-Value
Worst ES grade ≥ 3	3.72 (1.320–10.485)	0.013		
Pseudoglandular type	2.085 (1.102–3.944)	0.024		
Microvascular invasion *	5.197 (2.618–10.315)	<0.001	2.418 (1.146–5.099)	0.020
Satellite nodule	3.176 (1.503–6.710)	0.002		
Tumor necrosis	4.370 (2.236–8.543)	<0.001	2.592 (1.264–5.316)	0.009
Tumor hemorrhage	2.679 (1.407–5.102)	0.003		
Tumor size > 9 cm	5.896 (2.594–13.406)	<0.001		
Multiple tumors	2.628 (1.097–6.294)	0.030		
Beyond Milan criteria	4.350 (2.300–8.224)	<0.001	3.008 (1.548–5.843)	0.001
Serum AFP > 1000 IU/mL	3.170 (1.534–6.550)	0.002		

^1^ Abbreviations: EHR, extrahepatic recurrence; HR, hazard ratio; CI, confidence interval; ES, Edmonson–Steiner; HCC, Hepatocellular carcinoma; AFP, alpha-fetoprotein. * Confirmed at pathologic examination.

**Table 4 cancers-17-01417-t004:** Factors associated with overall survival after surgical resection of HCC.

Factors	Univariate Analysis		Multivariate Analysis	
	^1^ HR (95% CI)	*p*-Value	HR (95% CI)	*p*-Value
Child–Pugh class B	4.265 (1.035–17.573)	0.045		
MELD ≥ 10	3.701 (1.665–8.223)	0.001	3.315 (1.441–7.626)	0.005
Major ES grade ≥ 3	1.788 (1.058–3.022)	0.030		
Pseudoglandular type	2.006 (1.182–3.407)	0.010		
Microvascular invasion *	3.026 (1.592–5.751)	0.001		
Multicentricity	2.118 (1.032–4.347)	0.041		
Satellite nodule	2.212 (1.139–4.297)	0.019		
Tumor necrosis	2.530 (1.492–4.291)	0.001		
Tumor hemorrhage	1.917 (1.134–3.239)	0.015		
Tumor size > 9 cm	3.109 (1.326–7.289)	0.009		
Multiple tumors	4.438 (2.326–8.467)	<0.001	3.720 (1.879–7.365)	<0.001
Macrovascular invasion	2.300 (1.247–4.240)	0.008		
Beyond the Milan criteria	2.500 (1.448–4.314)	0.001		
Extrahepatic recurrence	14.044 (8.253–23.901)	<0.001	12.838 (7.324–22.503)	<0.001

^1^ Abbreviations: ES, Edmonson–Steiner; HR, hazard ratio; CI, confidence interval; HCC, Hepatocellular carcinoma. * Confirmed at pathologic examination.

## Data Availability

The datasets generated and/or analyzed during the current study are available from the corresponding author upon reasonable request.

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
