# Peer review of "Extrahepatic Recurrence After Surgical Resection of Hepatocellular Carcinoma Without Intrahepatic Recurrence: A Multi-Institutional Observational Study"

_cancers, 2025, doi:10.3390/cancers17091417_

Round 1
Reviewer 1 Report
Comments and Suggestions for Authors
This is an observational study on the extrahepatic recurrence (EHR) without intrahepatic recurrence (IHR) after first time curative resection for hepatocellular carcinoma (HCC). It involved 4 tertiary centers between 1/2004 and 12/2019. The result showed that the incidence of EHR without IHR was 6.7% (38 patients). EHR was associated with microvascular invasion, tumor necrosis and advanced stage beyond Milan criteria by multivariate analysis. EHR was also associated with poor survival.
There are some queries and comments for the authors.
- In the abstract, it was mentioned that the study included 1,069 treatment-naive patients who underwent curative hepatectomy but in the text and Fig. 1 it was 1066 patients. In the abstract and the text, 569 patients were included in the final analysis but in Fig. 1 it was 535 patients included in final analysis, the numbers are confusing, the authors need to check and correct.
- Did the authors collect information of ruptured HCC for hepatectomy as this was a significant factor for EHR in peritoneum ?
- Except for Milan criteria, the other two factors associated with EHR were microvascular invasion and tumor necrosis which were only known after surgery, thus these were not useful in clinical practice for patient selection for surgery.
- Macrovascular invasion is theoretically associated with more EHR as tumor cells shed to circulation lead to distant metastasis like lung and bone. Why it was not a factor but instead only microvascular invasion a factor associated with EHR?
- The authors concluded that "risk stratification approach based on these factors facilitates a personalized strategy for early detection and intervention, potentially improving overall survival". Actually once EHR for HCC occurs, the prognosis is very poor and treatment option is limited, how can the authors improve the survival by early detection and intervention. The authors need evidence to substantiate the statement. Furthermore, what imaging and how frequent do the authors propose to detect EHR after liver resection in high risk patients?
- It is good if the authors can present what types of treatment that can be offered to the 38 patients diagnosed with EHR and what are the treatment outcomes.
Author Response
[Comment 1] In the abstract, it was mentioned that the study included 1,069 treatment-naive patients who underwent curative hepatectomy but in the text and Fig. 1 it was 1066 patients. In the abstract and the text, 569 patients were included in the final analysis but in Fig. 1 it was 535 patients included in final analysis, the numbers are confusing, the authors need to check and correct.
[Response 1] We sincerely appreciate your thoughtful comment. We acknowledge that there were typographical errors regarding the number of enrolled patients. The number of patients depicted in Figure 1 is accurate. The value reported as “1,069” in the abstract has been corrected to “1,066,” which is consistent with the rest of the manuscript.
As illustrated in Figure 1 and detailed in Table 1, we compared the risk variables of EHR without IHR between two groups: “No recurrence (n=531)” and “EHR without IHR (n=38).” (The mentioned figure of 535 patients represents the sum of those who experienced either IHR and/or EHR following surgical resection of HCC.)
[Comment 2] Did the authors collect information of ruptured HCC for hepatectomy as this was a significant factor for EHR in peritoneum ?
[Response 2] Thank you for your insightful comment. As rupture of HCC itself may represent a significant risk factor for the development of EHR, patients whose initial diagnosis of HCC was associated with hepatic rupture were excluded at the initiation of the study. This information has been added to the “2. Materials and Methods, 2.1 Patients” section (page 2).
[Comment 3] Except for Milan criteria, the other two factors associated with EHR were microvascular invasion and tumor necrosis which were only known after surgery, thus these were not useful in clinical practice for patient selection for surgery.
[Response 3] We appreciate your valuable comment. The authors acknowledge that microvascular invasion and tumor necrosis can only be confirmed through pathological examination following surgical resection. However, given that surgical resection remains one of the most definitive curative treatments for HCC, particularly in its early stages, and that no established neoadjuvant treatment exists for patients at high risk of post-surgical recurrence, the authors believe that identifying post-surgical pathological features associated with the risk of EHR may provide meaningful insights for clinical practice. This limitation has been added to the last part of “Discussion” (page 13).
[Comment 4] Macrovascular invasion is theoretically associated with more EHR as tumor cells shed to circulation lead to distant metastasis like lung and bone. Why it was not a factor but instead only microvascular invasion a factor associated with EHR?
[Response 4] We appreciate your insightful comment. Macrovascular invasion is a well-established prognostic factor associated with poor outcomes following surgical resection, particularly in relation to post-operative recurrence and EHR. Although macrovascular invasion was more frequently observed in patients with EHR (6.6% vs. 13.2%, p = 0.130), statistical significance was not achieved in the univariate analysis of factors associated with EHR in the absence of IHR. We postulate that this result may be attributable to the small number of patients with macrovascular invasion, as well as the limited extent of venous tumor thrombosis (Vp1 and Vp2) included in our cohort. This likely reflects the fact that surgical resection is typically contraindicated in patients with advanced macrovascular invasion.
In contrast, although microvascular invasion is likewise recognized as a poor prognostic indicator, its presence is often difficult to detect preoperatively. In our study population, microvascular invasion was more prevalent than macrovascular invasion and was more frequently observed in patients with EHR. This finding suggests that microvascular invasion may play a more substantial role in the development of EHR in the absence of IHR, thereby serving as a significant risk factor in this context.
[Comment 5] The authors concluded that "risk stratification approach based on these factors facilitates a personalized strategy for early detection and intervention, potentially improving overall survival". Actually once EHR for HCC occurs, the prognosis is very poor and treatment option is limited, how can the authors improve the survival by early detection and intervention. The authors need evidence to substantiate the statement. Furthermore, what imaging and how frequent do the authors propose to detect EHR after liver resection in high risk patients?
[Response 5] We appreciate your insightful comment. While patients with EHR after treatment of HCC generally have limited therapeutic options beyond systemic therapy, potentially curative interventions such as metastasectomy or radiotherapy may be considered in select instances, particularly when EHR presents as a solitary lesion or involves a limited number of sites (oligometastasis), and have been linked to improve overall survival. However, in the majority of EHR cases, multiple metastases are more common than oligometastatic presentations, which contributes to less favorable clinical outcomes.
In this context, we aimed to identify potential risk factors associated with EHR in the absence of IHR, to facilitate early detection and timely intervention. Unfortunately, there remains no standardized imaging modality or surveillance interval specifically tailored for the detection of EHR following surgical resection of HCC. Nonetheless, in our cohort, 56.41% of EHR cases were identified using contrast-enhanced abdominal CT and 17.95% via chest CT. Based on these findings, we suggest—albeit with a low level of evidence—that in patients at high risk for EHR, regular surveillance using contrast-enhanced abdominal and chest CT, which are commonly employed in post-operative monitoring of HCC, may aid in the early detection of extrahepatic recurrence.
We have added this information at part of “Discussion” (page 13).
[Comment 6] It is good if the authors can present what types of treatment that can be offered to the 38 patients diagnosed with EHR and what are the treatment outcomes.
[Response 6] Thank you for your valuable comment. The information regarding treatment modalities and survival duration following the diagnosis of EHR in 38 patients has been incorporated into supplementary table 4, as detailed below.
Supplementary Table 4. Clinical findings of patients with EHR without intrahepatic HCC following surgical resection
Characteristic |
Patients without intrahepatic HCC at the time of EHR (n = 38) |
Neutrophil-to-lymphocyte ratio |
2.33 ± 2.20 |
Plt (×103/μL) |
170.56 ± 64.81 |
Serum AFP (IU/mL), median (range) |
22.78 (0.96–12,033.04) |
PIVKA (mAU/mL) |
120.00 (18.0–107,763.00) |
Recurrence-free survival, year (median range) |
1.07 (0.2–6.87) |
Location of metastasis, n (%) |
|
Lymph nodes |
7 (17.07) |
Bone |
7 (17.07) |
Lung |
15(36.59) |
Solitary/multiple |
4 (9.76)/11 (26.83) |
Peritoneum |
5 (12.20) |
Brain |
2 (4.88) |
Adrenal gl. |
5 (12.20) |
Diagnostic modality, n (%) |
|
Abdomen enhanced CT |
22 (56.41) |
Abdomen enhanced MRI |
1 (2.56) |
Chest X-ray |
2 (5.13) |
Chest enhanced CT |
7 (17.95) |
PET-CT |
4 (10.26) |
Spine MRI |
1 (2.56) |
Brain MRI |
2 (5.13) |
Survival duration after EHR (years) |
2.49 (0.098-10.80) |
Treatment modality after EHR |
|
Systemic chemotherapy |
7 (18.42) |
Radiotherapy |
15 (39.47) |
Metastatectomy |
3 (7.89) |
Combined therapy |
8 (21.05) |
Best supportive care |
1 (2.63) |
Follow-up loss |
4 (10.53) |
Reviewer 2 Report
Comments and Suggestions for Authors
Nice article.
Can you please discuss the role of biopsy of the HCC before transplantation to evaluate DNA-index and discuss favorable tumor biology.
Andreou A, Bahra M, Schmelzle M, Öllinger R, Sucher R, Sauer IM, Guel-Klein S, Struecker B, Eurich D, Klein F, Pascher A, Pratschke J, Seehofer D. Predictive factors for extrahepatic recurrence of hepatocellular carcinoma following liver transplantation. Clin Transplant. 2016 Jul;30(7):819-27. doi: 10.1111/ctr.12755.
Can you please discuss risk of immunosuppression and patient-related risk factors (age, tobacco, alcohol, etiology of liver disease) on extrahepatic malignancies.
Choudhary NS, Saigal S, Saraf N, Soin AS. Extrahepatic Malignancies and Liver Transplantation: Current Status. J Clin Exp Hepatol. 2021 Jul-Aug;11(4):494-500. doi: 10.1016/j.jceh.2020.10.008. Epub 2020 Oct 24. PMID: 34276155; PMCID: PMC8267344.
Author Response
Comment 1. Can you please discuss the role of biopsy of the HCC before transplantation to evaluate DNA-index and discuss favorable tumor biology.
Andreou A, Bahra M, Schmelzle M, Öllinger R, Sucher R, Sauer IM, Guel-Klein S, Struecker B, Eurich D, Klein F, Pascher A, Pratschke J, Seehofer D. Predictive factors for extrahepatic recurrence of hepatocellular carcinoma following liver transplantation. Clin Transplant. 2016 Jul;30(7):819-27. doi: 10.1111/ctr.12755.
Response 1. We thank the reviewer for the insightful comment.
As requested, we have added a discussion on the potential role of pre-transplant biopsy in evaluating tumor biology, including DNA ploidy and DNA index. These parameters have been explored as predictive factors for extrahepatic recurrence and overall prognosis following liver transplantation.
Specifically, we have revised the Discussion section to include the following sentence:
“In this context, the potential role of pre-transplant biopsy in evaluating tumor biology, such as DNA ploidy or DNA index, has also been highlighted as a way to improve risk stratification in LT candidates. Favorable tumor biology, as indicated by these molecular features, may help predict a lower risk of extrahepatic recurrence and refine selection criteria for transplantation.”
This addition follows the paragraph discussing the Milan criteria and the findings from Andreou et al., thereby placing the biopsy discussion within the broader context of tumor burden and transplant eligibility.
Comment 2. Can you please discuss risk of immunosuppression and patient-related risk factors (age, tobacco, alcohol, etiology of liver disease) on extrahepatic malignancies.
Choudhary NS, Saigal S, Saraf N, Soin AS. Extrahepatic Malignancies and Liver Transplantation: Current Status. J Clin Exp Hepatol. 2021 Jul-Aug;11(4):494-500. doi: 10.1016/j.jceh.2020.10.008. Epub 2020 Oct 24. PMID: 34276155; PMCID: PMC8267344.
Response 2. We thank the reviewer for the valuable comment.
As requested, we have added a discussion on the role of immunosuppression and patient-related risk factors in the development of de novo extrahepatic malignancies, particularly in the post-transplant population. This addition has been made in the Discussion section, immediately following the paragraph describing tumor markers and EHR detection methods.
The newly added text reads as follows:
“Furthermore, immunosuppression and patient-related factors such as age, tobacco and alcohol use, and the underlying etiology of liver disease have also been associated with an increased risk of de novo extrahepatic malignancies, particularly in the post-transplant population. These factors may compromise immune surveillance, facilitate oncogenic viral reactivation, and contribute to a proinflammatory microenvironment, thereby increasing malignancy risk and leading to poorer outcomes. As highlighted by Choudhary et al., identifying and managing these risk factors through tailored surveillance strategies can enhance early detection and potentially improve long-term prognosis in this high-risk cohort.”
This addition broadens the scope of our discussion and highlights the importance of personalized surveillance strategies beyond tumor biology alone, in accordance with the reviewer’s suggestion.
Reviewer 3 Report
Comments and Suggestions for Authors
Authors assessed the risk factors for EHR and developed the risk score for predicting EHR. Furthermore, authors developed the risk score for predicting OS. This study was interesting, and the manuscript was well-written. But several issues remained to be addressed.
- Authors showed the risk factor for EHR. As the variables, AFP>1000 was included. Authors should explain why the cut-off level was applied. Furthermore, authors should describe why PIVKA2 (DCP) was not included.
- In this study, most recurrence was found within 2 years. This observation might suggest the recurrence is already found at the resection. In this study, recurrence within 60 days after resection was excluded. Authors should describe the rationale of 60 days.
- In the prediction of EHR or OS, only tumor factors were involved. In clinical practice, liver reserve function is an essential factor in managing HCC, especially for OS. Systemic therapies can be done for patients with good hepatic reserve. Hepatic reserve function should be included to the variable, especially in OS.
Author Response
Comment 1. Authors showed the risk factor for EHR. As the variables, AFP>1000 was included. Authors should explain why the cut-off level was applied. Furthermore, authors should describe why PIVKA2 (DCP) was not included.
Response 1. We thank the reviewer for the valuable comment and the opportunity to clarify.
Elevated AFP levels are widely recognized as being associated with poor prognosis and aggressive tumor behavior in hepatocellular carcinoma. In our analysis, we evaluated multiple AFP thresholds and identified 1000 ng/mL as the value most strongly associated with extrahepatic recurrence. Therefore, AFP > 1000 ng/mL was selected as a binary variable for inclusion in the multivariate model.
Regarding PIVKA-II (DCP), although it is a well-established tumor marker for HCC, test results were not consistently available for a substantial proportion of our study cohort. Due to this limitation in data completeness, we excluded PIVKA-II from the multivariate analysis to preserve statistical reliability. This rationale has been added and the limitation has been acknowledged in the revised Discussion section, as follows:
“Among the tumor markers evaluated, various AFP thresholds were considered, and a cut-off value of 1000 ng/mL was selected based on its strongest association with EHR in our dataset. This threshold is also supported by previous studies reporting poor prognosis and increased risk of metastasis in patients with markedly elevated AFP levels.
PIVKA-II (DCP), although a well-known marker of HCC, was excluded from the multivariate analysis due to a substantial proportion of missing data, which limited its utility as a reliable variable in our statistical model and is acknowledged as a limitation of this study.”
Comment 2. In this study, most recurrence was found within 2 years. This observation might suggest the recurrence is already found at the resection. In this study, recurrence within 60 days after resection was excluded. Authors should describe the rationale of 60 days.
Response 2. We thank the reviewer for the insightful comment and the opportunity to clarify this methodological choice.
To minimize the inclusion of pre-existing but radiologically undetected extrahepatic metastases that may have already been present at the time of surgery, we excluded patients who developed EHR within 60 days after resection. While a longer exclusion interval (e.g., 6 to 12 months) could further improve specificity in identifying true postoperative recurrence, we adopted a 60-day threshold in consideration of the limited number of patients with EHR without IHR in our cohort. This cutoff was chosen to balance the risk of including residual disease with the need to maintain sufficient statistical power for meaningful analysis.
We have added a detailed explanation of this rationale in the Methods section and have acknowledged the limitation of this approach in the Discussion, suggesting that future large-scale studies with longer recurrence-free intervals (e.g., one year or longer) could better distinguish true postoperative recurrence from residual disease that may have been present at the time of surgery.
Comment 3. In the prediction of EHR or OS, only tumor factors were involved. In clinical practice, liver reserve function is an essential factor in managing HCC, especially for OS. Systemic therapies can be done for patients with good hepatic reserve. Hepatic reserve function should be included to the variable, especially in OS.
Response 3. We sincerely appreciate your insightful comment. In response, we evaluated the impact of MELD score, Child-Pugh score, and ALBI grade—each representing hepatic functional reserve—on the development of EHR and overall survival following surgical resection of HCC. None of these parameters demonstrated a statistically significant association with EHR development (MELD: HR = 1.033, p = 0.752; Child-Pugh: HR = 3.075, p = 0.268; ALBI: HR = 2.105, p = 0.078).
In contrast, when assessing overall survival, both Child-Pugh class B and MELD ≥ 10 were significantly associated with poor prognosis in univariate analysis (HR = 4.265, p = 0.045; and HR = 0.3701, p = 0.001, respectively). In multivariate analysis, MELD ≥ 10 remained an independent predictor of poor survival (HR = 3.315, p = 0.005).
In light of your valuable suggestion, we have revised Table 4 to include MELD ≥ 10 as a final prognostic factor for overall survival and have updated the corresponding section of the manuscript accordingly (page 12).
Round 2
Reviewer 1 Report
Comments and Suggestions for Authors
Thanks for replying the questions raised and addressing the limitations of this study.
Author Response
Comments : Thanks for replying the questions raised and addressing the limitations of this study.
Response : We thank the reviewer for the encouraging comment.
Reviewer 3 Report
Comments and Suggestions for Authors
Revised manuscript was well-addressed to the reviewer's comments. The manuscript has been improved the significance. Old table 4 was present in the revised manuscript. It should be corrected. Authors should perform ROC analysis to determine optimal cut-off level in AFP.
Author Response
Comments:
Revised manuscript was well-addressed to the reviewer's comments. The manuscript has been improved the significance. Old table 4 was present in the revised manuscript. It should be corrected. Authors should perform ROC analysis to determine optimal cut-off level in AFP.
Response 1:
We greatly appreciate the reviewer’s valuable comment and careful review.First, regarding Table 4, we would like to clarify that the updated version had been included in our revised submission. However, it appears that the previous version may have been inadvertently forwarded during the review process. We fully understand how this might have caused confusion and sincerely appreciate the reviewer for pointing it out. We have confirmed that the current version of Table 4 in the submission system is now correct and up to date.
Second, in accordance with the reviewer’s suggestion, we performed a receiver operating characteristic (ROC) analysis to determine the optimal cut-off level of AFP for predicting extrahepatic recurrence (EHR). The analysis showed that the optimal AFP value was between 997 and 1,011 ng/mL (AUC = 0.591, Youden index = 0.183). Based on these findings and for clinical practicality, we selected 1,000 ng/mL as the final cut-off value.
This information has been incorporated into the revised Discussion section as follows:
“Among the tumor markers evaluated, various AFP thresholds were considered, and a cut-off value of 1,000 ng/mL was selected based on its strongest association with EHR in our dataset (AUC = 0.591). This threshold is also supported by previous studies reporting poor prognosis and increased risk of metastasis in patients with markedly elevated AFP levels.”
We thank the reviewer once again for the helpful recommendations that have further enhanced the clarity and robustness of our manuscript.